# Association of Estimated Glomerular Filtration Rate with Risk of Head and Neck Cancer: A Nationwide Population-Based Study

**DOI:** 10.3390/cancers14204976

**Published:** 2022-10-11

**Authors:** Hyun-Bum Kim, Jun-Ook Park, Inn-Chul Nam, Choung-Soo Kim, Sung Joon Park, Dong-Hyun Lee, Kyungdo Han, Young-Hoon Joo

**Affiliations:** 1Department of Otolaryngology-Head and Neck Surgery, College of Medicine, The Catholic University of Korea, Seoul 06591, Korea; 2Department of Otolaryngology-Head and Neck Surgery, Chung-Ang University College of Medicine, Chung-Ang University Gwangmyeong Hospital, Gwangmyeon-si 14353, Korea; 3Department of Statistics and Actuarial Science, Soongsil University, Seoul 06978, Korea

**Keywords:** head and neck neoplasms, glomerular filtration rate, kidney disease, epidemiology, Korea

## Abstract

**Simple Summary:**

A decreased estimated glomerular filtration rate is associated with several types of cancer. However, there are controversies regarding such an association between the estimated glomerular filtration rate and head and neck cancer. This is an observational cohort study using data from the Korean national health claims database. Elevated estimated glomerular filtration rate was associated with a risk of head and neck cancer incidence.

**Abstract:**

In this study, through a cohort study of 10 million people, we investigated the association between estimated glomerular filtration rate (eGFR) and head and neck cancer (HNC) incidence. This is an observational cohort study using data from the national health claims database established by the Korean National Health Insurance Service (NHIS). We selected 9,598,085 participants older than 20 years who had undergone health checkups in 2009. A health checkup involves the history of any diseases, current health status, and results of several physical and blood exams including eGFR. We investigated the presence of HNC diagnosis in their national health insurance data from 2010 to 2018. Of the 9,598,085 participants, 10,732 had been newly diagnosed with HNC in the 9-year follow-up. In the multivariate Cox proportional hazard model, participants with elevated eGFR were associated with a risk of HNC incidence (HR = 1.129; 95% CI = 1.075–1.186 for eGFR = 90–104 mL/min/1.73 m^2^ and HR = 1.129; 95% CI = 1.076–1.194 for eGFR ≥ 105 mL/min/1.73 m^2^) compared with those with eGFR 60–89 mL/min/1.73 m^2^. Among HNC, the incidences of oral cavity, oropharyngeal, hypopharyngeal, and laryngeal cancers were significantly increased in the elevated eGFR group. According to the subgroup analysis, participants with eGFR ≥ 60 mL/min/1.73 m^2^ were correlated with risk of HNC incidence in middle age, non/mild drinker, low BMI, no diabetes, and no hypertension patients compared with those with eGFR < 60 mL/min/1.73 m^2^. Elevated eGFR was associated with the risk of some type of HNC, even in individuals with adjusted hypertension and diabetes without chronic diseases. The results of this study have implications for etiological investigations and preventive strategies.

## 1. Introduction

Several factors affect the incidence of head and neck cancer (HNC). Genetic factors such as gender and age, as well as environmental factors such as smoking, alcohol drinking, and human papillomavirus are widely known [1]. Although it has been suggested that chronic diseases such as hypertension, diabetes, and dyslipidemia can increase the incidence of cancer, the pathologic mechanisms of chronic diseases causing cancer are too diverse and the duration is too long to form a strong conclusion. However, with the development of technology, many substances from cells have been identified microscopically, and various mechanisms have been elucidated. Macroscopically, through various big data studies, it has been confirmed that the incidence of cancer is increasing in patients with chronic diseases [2].

Chronic kidney disease (CKD) is defined as a persistent abnormality in kidney structure and function for more than three months. Equations estimating the glomerular filtration rate (eGFR) are important clinical tools in detecting and managing kidney disease. The most important criteria for judging kidney function are eGFR less than 60 mL/min/1.73 m^2^ and albuminuria of at least 30 mg per 24 h or a urine albumin-to-creatinine ratio of at least 30 mg/g [3]. It is well known that decreased eGFR increases the incidence of coronary artery disease, myocardial infarction, heart failure, atrial fibrillation, and cerebrovascular events [3,4]. Decreased eGFR is also associated with several types of cancer [5,6]. In some studies, decreased eGFR increases the incidence of liver, bladder, kidney, and renal cancers [6,7,8].

Elevated eGFR (glomerular hyperfiltration) is a functional abnormality in insulin-dependent diabetes mellitus and is associated with cancer development, mortality risk, and early kidney disease [9,10]. Hyperfiltration is hypothesized to be a precursor of intraglomerular hypertension leading to albuminuria [10]. eGFR progressively decreases in parallel with a further increase in albuminuria, which ultimately can lead to end-stage renal failure. In this study, through a cohort study of 10 million people, we investigated the association between eGFR and HNC incidence using data from a nationwide population-based study.

## 2. Materials and Methods

### 2.1. Study Population

This observational cohort study obtained data from the national health claims database established by the Korean National Health Insurance Service (NHIS). The Korean NHIS is the public medical insurance system, which is administered by the Ministry for Health, Welfare, and Family Affairs [11]. The computerized database of the NHIS includes all claims data containing medical information related to patients. Diagnoses were confirmed using the International Classification of Disease, Tenth Revision, Clinical Modification (ICD-10) codes C02, C03, C04, C05, and C06 for oral cavity cancer; C07 and C08 for salivary gland cancer; C11 for nasopharyngeal cancer; C01, C051, C099, and C103 for oropharyngeal cancer; C12 and C13 for hypopharyngeal cancer; C10 for sinonasal cancer; and C32 for laryngeal cancer. The present study protocol was reviewed and approved by the Institutional Review Board (IRB)/Ethics Committee approval for the study was obtained from The Catholic University of Korea in accordance with the Declaration of Helsinki. [SC21ZISE0024].

### 2.2. Patient Selection

Enrollees in the National Health Insurance Corporation are recommended to undergo a standardized medical examination every two years. We selected subjects with inclusion criteria of those who were older than 20 years and had undergone health checkups in 2009 (n = 10,585,852). We monitored the subjects until 31 December 2018. As exclusion criteria, we set missing data (n = 746,403), previous cancer history (n = 153,456), and individuals who were diagnosed with cancer within a one-year lag period to minimize detection bias (n = 87,908). Finally, 9,598,085 subjects were included in this study. Participants were defined as having HNC if they had admission records for HNC in their national health insurance data from 2010 to 2018. The medical examinations included measurements of height, weight, and blood pressure. In addition, the levels of fasting plasma glucose, triglycerides, total cholesterol, and HDL were obtained. Smoking criteria are divided into three classifications based on past and present smoking. Alcohol intake criteria are divided into three classifications based on frequency in 1 week and amount on one occasion (none; mild, <30 g of alcohol/day; heavy, ≥30 g of alcohol/day). The physical activity level was collected using standardized self-reporting International Physical Activity Questionnaire. Hypertension was defined as (1) one or more claims/year for an antihypertensive prescription under ICD-10 codes I10–I15 or (2) systolic/diastolic BP  ≥ 140/90 mmHg L without a claim for anti-hypertension medication under ICD-10 codes I10–13 and I15. Dyslipidemia was defined as (1) a fasting blood glucose level ≥ 126 mg/dL (≥7 mmol/L) or (2) the presence of one or more claims per year for antihyperglycemic medications with ICD-10-CM code E10-14.

We categorized participants by eGFR into five categories according to KDIGO (Kidney Disease: Improving Global Outcomes): increased for ≥105, normal for 104–90, mildly decreased for 60–89, moderately decreased for 30–59, and severely decreased for <30 mL/min/1.73 m^2^ [12]. The eGFR was estimated using the Modification of Diet in Renal Disease Study equation (eGFR = 175 × [serum creatinine in mg/dL]^−1.154^ × [age]^−0.203^ × [0.742 in women]) [11].

### 2.3. Statistical Analysis

Statistical analyses were performed using SAS version 9.2 (SAS Institute, Cary, NC, USA). Basic characteristics are presented using descriptive analysis. Differences in baseline characteristics between groups were determined using Student’s t-test for continuous variables and the X^2^ test for categorical variables. HNC incidence was calculated by dividing the number of cases by 1000 person-years. Cox proportional hazards models were applied to estimate hazard ratios (HRs) and 95% confidence intervals (CIs) for the associations between eGFR and risk of HNC. Subgroup analyses were performed by multivariable Cox proportional hazard models. Model 1 was unadjusted. Model 2 was adjusted for age, sex, income, smoking, alcohol consumption, and regular exercise. Model 3 was adjusted for age, sex, income, smoking, alcohol consumption, regular exercise, diabetes, and hypertension. A *p*-value < 0.05 was considered statistically significant.

## 3. Results

### 3.1. General Characteristics

The characteristics of the study population are presented in Table 1. We identified 658,550 participants with eGFR < 60 mL/min/1.73 m^2^, who were more likely to be older and women and exhibited a higher prevalence of non/ex-smokers and non/mild drinkers. Comorbidities of hypertension, diabetes, dyslipidemia, high BMI, and high total cholesterol were observed more frequently in the eGFR < 60 mL/min/1.73 m^2^ group than the eGFR ≥ 60 mL/min/1.73 m^2^ group (all, *p* < 0.0001). The mean eGFR values in the eGFR < 60 mL/min/1.73 m^2^ group and eGFR ≥ 60 mL/min/1.73 m^2^ group were 37.02 ± 23.09 and 91.36 ± 44.27 mL/min/1.73 m^2^, respectively.

### 3.2. Association between eGFR and HNC

Among the 9,598,085 participants, 10,732 were newly diagnosed as HNC. The unadjusted and multivariable-adjusted HRs of HNC according to the presence or absence of CKD are presented in Table 2. Among the data, adjustment was performed based on factors and diseases that affected eGFR. Age, gender, income, smoking status, alcohol intake, exercise, diabetes, and hypertension-adjusted hazard ratios indicate that participants with high eGFR were associated with risk of HNC compared with those with eGFR 60–89 mL/min/1.73 m^2^ (HR = 1.129; 95% CI = 1.075–1.186 for eGFR = 90–104 mL/min/1.73 m^2^ and HR = 1.129; 95% CI = 1.076–1.194 for eGFR ≥ 105 mL/min/1.73 m^2^). Among head and neck cancers, the incidences of oral cavity cancer (HR = 1.149; 95% CI = 1.032–1.28 for eGFR = 90–104 mL/min/1.73 m^2^), oropharyngeal cancer (HR = 1.233; 95% CI = 1.08–1.408 for eGFR ≥ 105 mL/min/1.73 m^2^), hypopharyngeal cancer (HR = 1.392; 95% CI 1.182–1.639 for eGFR = 90–104 mL/min/1.73 m^2^ and HR = 1.527; 95% CI = 1.274–1.831 for eGFR ≥ 105 mL/min/1.73 m^2^), and laryngeal cancer (HR = 1.167; 95% CI 1.063–1.282 for eGFR = 90–104 mL/min/1.73 m^2^ and HR = 1.148; 95% CI = 1.028–1.282 for eGFR ≥ 105 mL/min/1.73 m^2^) were significantly increased in the high eGFR group compared with the eGFR 60–89 mL/min/1.73 m^2^ group. These results are also confirmed graphically in Figure 1.

### 3.3. Subgroup Analysis

We also conducted comparisons of some items between groups with eGFR < 60 mL/min/1.73 m^2^ and eGFR ≥ 60 mL/min/1.73 m^2^ (Table 3). In multivariate logistic analysis, the middle-aged group from 41 to 64 years with eGFR ≥ 60 mL/min/1.73 m^2^ was correlated with higher risk of HNC compared with that with eGFR < 60 mL/min/1.73 m^2^ (HR = 1.133; 95% CI = 1.022–1.256). Elevated risk of HNC was observed in eGFR ≥ 60 mL/min/1.73 m^2^ participants with non/mild drinker status (HR = 1.082; 95% CI = 1.007–1.162), low BMI (HR = 1.098; 95% CI = 1.008–1.196), absence of diabetes (HR = 1.118; 95% CI = 1.034–1.208), and absence of hypertension (HR = 1.116; 95% CI = 1.005–1.204).

## 4. Discussion

We present a data-driven approach to identify the models that most appropriately describe the association between eGFR and risk of HNC in representative sample of the Korean population. In this study, participants with elevated eGFR ≥ 90 mL/min/1.73 m^2^ were associated with higher risk of HNC compared with those with eGFR 60–89 mL/min/1.73 m^2^. Our results support elevated GFR (glomerular hyperfiltration) as potentially reflecting renal injury [13]. Many previous studies have reported similar results, suggesting that higher eGFR may increase the incidence of cancer. Lowrance et al. reported that the incidence of any cancer is lower in 60–89 mL/min/1.73 m^2^ eGFR and 45–59 mL/min/1.73 m^2^ eGFR than in 90–150 mL/min/1.73 m^2^ eGFR [8]. Xu et al. reported that the incidence of any cancer is lower in 90–104 mL/min/1.73 m^2^ eGFR compared with eGFR greater than 105 mL/min/1.73 m^2^ [14]. They reported a U-shaped association between eGFR and overall cancer risk. Mok et al. reported the lowest overall cancer incidence in patients with 45–59 mL/min/1.73 m^2^ eGFR [15]. The incidence of overall cancer is higher in those with eGFR greater than 90 mL/min/1.73 m^2^ than in those with eGFR 60–89 mL/min/1.73 m^2^ or 45–59 mL/min/1.73 m^2^. They reported that the incidence of overall cancer exhibits a general J-shaped association with eGFR. In addition, they reported no association between laryngeal cancer and CKD. Considering that HNC is one of the main cancers of focus, the result of this study is consistent with the previous studies. A plausible explanation for these associations may be related to the damage caused by elevated GFR to the capacity of the renal tubules to reabsorb fluids and minerals from urine [16]. Damaged or destroyed tubules can lead to a common type of kidney injury known as acute tubular necrosis, which has been implicated in kidney failure [10]. In subgroup analysis, the high eGFR ≥ 60 mL/min/1.73 m^2^ group was associated with incidence of HNC in participants with middle age, low BMI, no diabetes, or no hypertension. This finding emphasizes that people with adjusted hypertension and diabetes without chronic disease should be aware of the possibility of HNC if eGFR is 60 mL/min/1.73 m^2^ or higher.

In this study, there was no significant relationship between decreased eGFR and risk of cancer. This is not fully explained, but the reasons are inferred as follows. First, several studies have reported that CKD is only a risk factor for cancer at specific sites such as the urinary tract, lung, liver, prostate, and breast [6,7,8,17,18]. Excluding these kinds of cancer, the general conclusion of the studies is that lower eGFR is not a risk factor for other kinds of cancer. This reasoning requires more research, but some studies revealed that strong carcinogens causing renal, pelvis, ureter, and bladder cancers such as aristolochic acid could induce interstitial nephritis or ESRD [7,19]. Up to this point, no substance has influenced both HNC and chronic inflammation of the kidney. Second, creatine itself is a substance mainly separated from muscle tissue. Therefore, it is greatly affected by gender, age, and obesity [20]. For this reason, its values are typically less accurate in women than in men. As a result, in women, eGFR is a less accurate representation of kidney status, and a study on its relationship with cancer may be inaccurate [18]. In addition, treatment using angiotensin-converting enzyme inhibitors (ACEi) or angiotensin II receptor blockers (ARB) are typical in CKD patients [3]. These agents remain controversial for use against lung cancer, but several studies have indicated that they reduce the incidence of some types of cancer [21,22]. The effect of these agents on the incidence of HNC should be further studied.

There are some limitations to this study. First, we only performed the study with eGFR and did not perform urinalysis. Albuminuria is one of the most important evaluation factors in CKD and plays an important role in severity [3]. However, because this study used big data, information about urinalysis could not be obtained. Accordingly, we set the cut-off of eGFR at 60 mL/min/1.73 m^2^ because values below 60 mL/min/1.73 m^2^ represent moderate, high, and very-high risk groups regardless of proteinuria [3]. Second, additional studies on mortality should be conducted. Although it did not affect the incidence of HNC, CKD may have a significant influence on the treatment results of HNC. Surgery on HNC itself is dangerous, and free flap reconstruction is occasionally necessary and is affected greatly by general and vessel conditions. In addition, in advanced stages of HNC, almost all patients undergo chemotherapy with agents such as cisplatin [23]. Cisplatin is part of the platinum-based antineoplastic family and is widely used as a representative agent in HNC treatment [24,25]. Cisplatin can cause acute kidney injury (AKI) or CKD, increasing patient mortality [26]. Hyperkalemia, metabolic acidosis, hyperphosphatemia, vitamin D deficiency, secondary hyperparathyroidism, and anemia-induced CKD can also increase mortality from HNC [10,27]. Third, dialysis and kidney transplantation have become more popular options for treatment in recent years but were not considered in this study. Vajdic et al. reported that the incidence of oral cavity cancer, not of all subtypes of HNC, increases during dialysis and after kidney transplantation [28]. In our findings, there was no information on dialysis or recommendation for kidney transplantation or not. Finally, eGFR was measured only once in a health checkup in 2009. Dynamic changes of eGFR could affect the incidence of HNC. Although these limitations exist, this study is meaningful in that it is the first to directly investigate the relationship between eGFR and HNC.

## 5. Conclusions

In our large-scale nationwide Korean cohort, we have demonstrated that elevated eGFR is a risk factor for several types of HNC such as oral cavity, oropharyngeal, hypopharyngeal, and laryngeal cancers. The present data provide a solid basis for future, larger studies aimed to assess whether eGFR screening of healthy people with adjusted hypertension and diabetes may significantly decrease HNC incidence and mortality risk to justify the cost and effort.

## Figures and Tables

**Figure 1 cancers-14-04976-f001:**
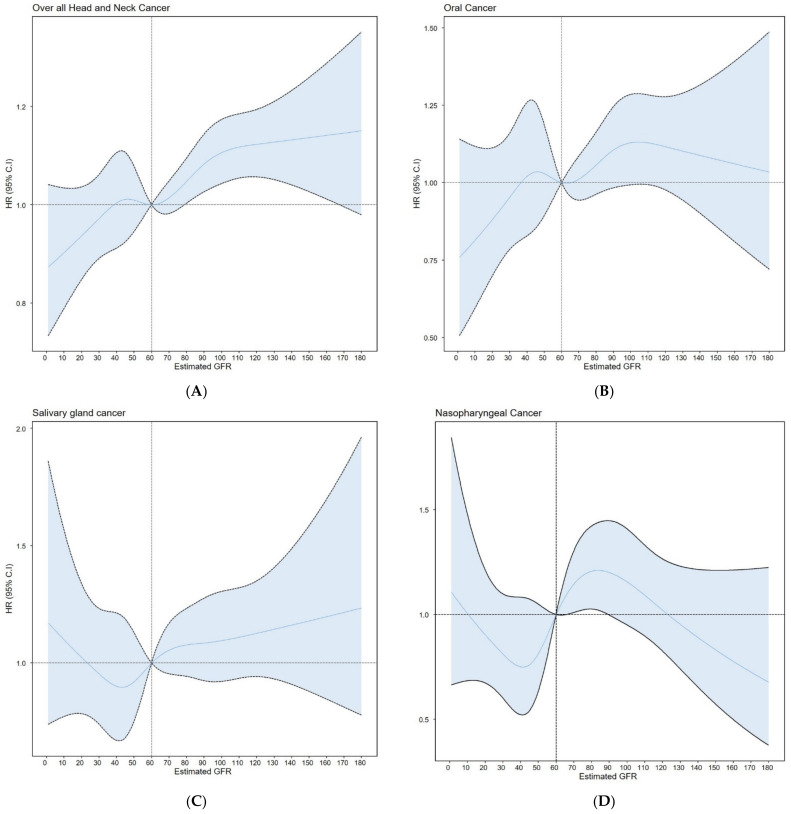
A restricted cubic spline showed the relationship between the estimated glomerular filtration rate and subtypes of head and neck cancer: (**A**) All head-and-neck cancers; (**B**) oral cancer, (**C**) salivary gland cancer, (**D**) nasopharyngeal cancer, (**E**) oropharyngeal cancer, (**F**) hypopharyngeal cancer, (**G**) sinonasal cancer, and (**H**) laryngeal cancer.

**Table 1 cancers-14-04976-t001:** Analysis of factors potentially associated with estimated glomerular filtration rate (n = 9,598,085).

Parameter	Low eGFR (<60 mL/min/1.73 m^2^) (n = 658,550)	High eGFR (≥60 mL/min/1.73 m^2^) (n = 8,939,535)	*p*-Value
Age (years)			<0.0001 *
<40	120,242 (18.26%)	2,909,956 (32.55%)	
40–64	319,839 (48.57%)	5,023,229 (56.19%)	
≥65	218,469 (33.17%)	1,006,350 (11.26%)	
Gender			<0.0001 *
Male	314,306 (47.73%)	4,906,495 (54.89%)	
Female	344,244 (52.27%)	4,033,040 (45.11%)	
Smoking status			<0.0001 *
Non-smoker	436,440 (66.27%)	5,335,922 (59.69%)	
Ex-smoker	98,814 (15%)	1,226,638 (13.72%)	
Current smoker	123,296 (18.72%)	2,376,975 (26.59%)	
Drinking status			<0.0001 *
Non-drinker	404,933 (61.49%)	4,534,378 (50.72%)	
Mild drinker	218,057 (33.11%)	3,674,516 (41.1%)	
Heavy drinker	35,560 (5.4%)	730,641 (8.17%)	
Regular exercise	127,064 (19.29%)	1,581,348 (17.69%)	<0.0001 *
Income (Q1)	112,252 (17.05%)	1,765,664 (19.75%)	<0.0001 *
Diabetes	104,538 (15.87%)	726,101 (8.12%)	<0.0001 *
Hypertension	279,773 (42.48%)	2,187,071 (24.47%)	<0.0001 *
Dyslipidemia	185,086 (28.11%)	1,551,280 (17.35%)	<0.0001 *
Body mass index (kg/m^2^)	24.07 ± 3.23	23.67 ± 3.47	<0.0001 *
HDL cholesterol (mg/dL)	56.13 ± 28.72	61.41 ± 66.45	<0.0001 *
LDL cholesterol (mg/dL)	118.63 ± 88.32	120.52 ± 207.77	<0.0001 *
Total cholesterol (mg/dL)	199.07 ± 43.82	195.07 ± 41.28	<0.0001 *
Triglyceride (mg/dL)	124.14 (123.97–124.31)	111.87(111.83–111.92)	<0.0001 *
GFR, mL/min/1.73 m^2^	37.02 ± 23.09	91.36 ± 44.27	<0.0001 *

Values are mean ± SE or % ± SE. * Significant at *p* < 0.05.

**Table 2 cancers-14-04976-t002:** Hazard ratios of head and neck cancer and its subtypes according to the estimated glomerular filtration rate.

eGFR, mL/min/1.73 m^2^	N	Event	Duration	Incidence Rates	Hazard Ratio (95% Confidence Interval)					
					Model 1	*p*-Value	Model 2	*p*-Value	Model 3	*p*-Value
	Head and neck cancer									
<30	234,880	189	1,943,406	0.0972	0.685 (0.593–0.792)	<0.0001	0.87 (0.753–1.006)	<0.0001	0.866 (0.749–1.002)	<0.0001
30–59	423,670	756	3,355,373	0.2253	1.59 (1.474–1.714)		0.995 (0.921–1.075)		0.971 (0.899–1.05)	
60–89	5,142,935	6000	42,323,127	0.1417	1 (reference)		1 (reference)		1 (reference)	
90–104	2,150,487	2222	17,763,186	0.1250	0.883 (0.841–0.927)		1.128 (1.073–1.185)		1.129 (1.075–1.186)	
≥105	1,646,113	1565	13,577,801	0.1152	0.813 (0.769–0.86)		1.13 (1.069–1.196)		1.129 (1.067–1.194)	
	Oral cavity cancer									
<30	234,880	34	1,943,932	0.0174	0.594 (0.423–0.836)	<0.0001	0.754 (0.536–1.06)	0.0419	0.75 (0.533–1.055)	0.0335
30–59	423,670	166	3,357,117	0.0494	1.689 (1.436–1.986)		1.008 (0.854–1.19)		0.981(0.831–1.158)	
60–89	5,142,935	1243	42,337,883	0.0293	1 (reference)		1 (reference)		1 (reference)	
90–104	2,150,487	465	17,768,720	0.0261	0.892 (0.802–0.992)		1.147 (1.03–1.278)		1.149 (1.032–1.28)	
≥105	1,646,113	317	13,581,595	0.0233	0.796(0.703–0.9)		1.063 (0.939–1.204)		1.062 (0.937–1.203)	
	Salivary gland cancer									
<30	234,880	28	1,943,895	0.0144	0.848 (0.581–1.237)	0.0004	1.035 (0.709–1.511)	0.8702	1.032 (0.707–1.507)	0.8275
30–59	423,670	79	3,357,315	0.0235	1.388 (1.1–1.751)		0.9 (0.71–1.14)		0.886 (0.699–1.123)	
60–89	5,142,935	719	42,339,039	0.0169	1 (reference)		1 (reference)		1 (reference)	
90–104	2,150,487	254	17,769,084	0.0142	0.842 (0.729–0.971)		1.027 (0.889–1.187)		1.028 (0.889–1.188)	
≥105	1,646,113	193	13,581,887	0.0142	0.837 (0.714–0.981)		1.04 (0.886–1.221)		1.038 (0.884–1.219)	
	Nasopharyngeal cancer									
<30	234,880	24	1,943,924	0.0123	0.814 (0.542–1.224)	0.0048	0.892 (0.593–1.342)	0.1977	0.892 (0.593–1.342)	0.2197
30–59	423,670	49	3,357,345	0.0145	0.959 (0.717–1.282)		0.746 (0.555–1.002)		0.753 (0.56–1.012)	
60–89	5,142,935	642	42,339,312	0.0151	1 (reference)		1 (reference)		1 (reference)	
90–104	2,150,487	240	17,769,217	0.0135	0.891 (0.769–1.034)		1.038 (0.893–1.206)		1.036 (0.891–1.203)	
≥105	1,646,113	146	13,582,025	0.0107	0.709 (0.593–0.849)		0.894 (0.74–1.072)		0.893 (0.745–1.07)	
	Oropharyngeal cancer									
<30	234,880	37	1,943,913	0.019	0.792 (0.571–1.1)	<0.0001	0.962(0.692–1.337)	0.0392	0.959 (0.69–1.332)	0.0398
30–59	423,670	121	3,357,170	0.036	1.507 (1.248–1.82)		1.021 (0.843–1.238)		1.003 (0.827–1.216)	
60–89	5,142,935	1014	42,338,430	0.0239	1 (reference)		1 (reference)		1 (reference)	
90–104	2,150,487	354	17,768,915	0.0199	0.832 (0.737–0.939)		1.04 (0.92–1.175)		1.041 (0.921–1.176)	
≥105	1,646,113	288	13,581,616	0.0212	0.886 (0.777–1.01)		1.235 (1.082–1.41)		1.233 (1.08–1.408)	
	Hypopharyngeal cancer									
<30	234,880	6	1,943,996	0.003	0.273 (0.122–0.61)	<0.0001	0.415 (0.186–0.929)	<0.0001	0.411 (0.183–0.919)	<0.0001
30–59	423,670	72	3,357,399	0.0214	1.888 (1.474–2.419)		1.119 (0.87–1.439)		1.068 (0.83–1.375)	
60–89	5,142,935	479	42,340,319	0.0113	1 (reference)		1 (reference)		1 (reference)	
90–104	2,150,487	212	17,769,548	0.0119	1.056 (0.898–1.241)		1.386(1.177–1.633)		1.392 (1.182–1.639)	
≥105	1,646,113	160	13,582,108	0.0117	1.042 (0.871–1.246)		1.527(1.273–1.831)		1.527 (1.274–1.831)	
	Sinonasal cancer									
<30	234,880	5	1,943,991	0.0025	0.344 (0.142–0.832)	0.0001	0.435 (0.18–1.054)	0.4636	0.434 (0.179–1.05)	0.4335
30–59	423,670	38	3,357,440	0.0113	1.511 (1.079–2.115)		0.928 (0.659–1.309)		0.901 (0.638–1.271)	
60–89	5,142,935	317	42,340,560	0.0074	1 (reference)		1 (reference)		1 (reference)	
90–104	2,150,487	105	17,769,706	0.0059	0.79 (0.633–0.985)		1.003 (0.802–1.254)		1.005 (0.804–1.257)	
≥105	1,646,113	74	13,582,309	0.0054	0.728 (0.565–0.937)		0.967 (0.749–1.248)		0.965 (0.747–1.246)	
	Laryngeal cancer									
<30	234,880	55	1,943,774	0.0282	0.719 (0.55–0.941)	<0.0001	0.995 (0.76–1.302)	0.0092	0.987 (0.754–1.293)	0.0084
30–59	423,670	237	3,356,758	0.0706	1.794 (1.566–2.056)		1.087 (0.946–1.248)		1.041 (0.906–1.196)	
60–89	5,142,935	1665	42,335,842	0.0393	1 (reference)		1 (reference)		1 (reference)	
90–104	2,150,487	612	17,768,002	0.3444	0.876 (0.799–0.961)		1.163 (1.059–1.277)		1.167 (1.063–1.282)	
≥105	1,646,113	403	13,581,165	0.0296	0.754 (0.677–0.841)		1.147 (1.028–1.281)		1.148 (1.028–1.282)	

Model 1: Unadjusted; Model 2: Age, sex, income, smoking, alcohol consumption, and regular exercise; Model 3: Age, sex, income, smoking, alcohol consumption, regular exercise, diabetes, and hypertension.

**Table 3 cancers-14-04976-t003:** Analysis of factors potentially associated with head and neck cancer according to the estimated glomerular filtration rate.

Parameter	eGFR	Number	Event	Duration	Rates	HR (95% CI)
Age (years)						
20–40	Low	120,242	24	1,009,741	0.0237	1 (reference)
	High	2,909,956	720	24,139,339	0.0298	1.228 (0.818–1.846)
41–64	Low	319,839	388	2,645,875	0.1466	1 (reference)
	High	5,023,229	6159	41,622,530	0.1479	1.133 (1.022–1.256)
≥65	Low	218,469	533	1,643,161	0.3243	1 (reference)
	High	1,006,350	2908	7,902,245	0.368	0.915 (0.832–1.005)
Gender						
Male	Low	314,306	697	2,517,359	0.2768	1 (reference)
	High	4,906,495	7803	40,213,973	0.194	1.080 (0.998–1.168)
Female	Low	344,244	248	2,781,420	0.0891	1 (reference)
	High	4,033,040	1984	33,450,142	0.0593	1.028 (0.897–1.178)
Smoking status						
Never or Ex-smoker	Low	535,254	652	4,305,152	0.1514	1 (reference)
	High	6,562,560	5860	54,198,226	0.1081	1.076(0.991–1.169)
Current smoker	Low	123,296	293	993,626	0.2948	1 (reference)
	High	2,376,975	3927	19,465,888	0.2017	1.082 (0.959–1.220)
Alcohol intake						
<30 g/day	Low	622,990	860	5,009,991	0.1716	1 (reference)
	High	8,208,894	8307	67,681,338	0.1227	1.082 (1.007–1.162)
≥30 g/day	Low	35,560	85	288,788	0.2943	1 (reference)
	High	730,641	1480	5,982,776	0.2473	1.186 (0.951–1.477)
Body mass index						
<25 kg/m^2^	Low	414,523	595	3,323,155	0.179	1 (reference)
	High	6,051,975	6652	49,826,663	0.1335	1.098 (1.008–1.196)
≥25 kg/m^2^	Low	244,027	350	1,975,624	0.1771	1 (reference)
	High	2,887,560	3135	23,837,452	0.1315	1.093 (0.976–1.223)
Diabetes						
Yes	Low	104,538	233	792,792	0.2939	1 (reference)
	High	726,101	1495	5,846,361	0.2557	0.993 (0.862, 1.145)
No	Low	554,012	712	4,505,986	0.158	1 (reference)
	High	8,213,434	8292	67,817,753	0.1222	1.118 (1.034–1.208)
Hypertension						
Yes	Low	279,773	566	2,174,976	0.2602	1 (reference)
	High	2,187,071	4052	17,793,001	0.2277	1.039 (0.950–1.137)
No	Low	378,777	379	3,123,803	0.1213	1 (reference)
	High	6,752,464	5735	55,871,114	0.1026	1.116 (1.005–1.240)

Low eGFR: <60 mL/min/1.73 m^2^, high eGFR: ≥60 mL/min/1.73 m^2^; Adjusted for age, sex, income, smoking, alcohol consumption, regular exercise, diabetes, and hypertension.

## Data Availability

Data are available on request due to data sharing restrictions. The data presented in this study are available on request from the corresponding author.

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
