# Peer review of "Association of Estimated Glomerular Filtration Rate with Risk of Head and Neck Cancer: A Nationwide Population-Based Study"

_cancers, 2022, doi:10.3390/cancers14204976_

Round 1
Reviewer 1 Report
This study explored the relationship between glomerular filtration rate and risk of head and neck cancer using data from a national health claims database. This is a study with large size. The author found Elevated eGFR was associated with risk of some type of HNC, even in healthy individuals without chronic diseases. This is an interesting study.
I have several suggestions you may considered.
1.There is no criteria of inclusion and exclusion for patients’ selection.
2. The author should list a subhead: data collection with details about how they collect the data. Some definition for some variables were not given such as smoking, drinking, physical activity. The author should give specific standard.
3. I suggest the author may draw a restricted cubic spline that showed the relationship between eGFR and risk of HNSCC.
4.Statistialanalysis: the author should present more details for statistical method descriptions. How you deal with variable? How do you perform the analysis? How do you define the model 1, model 2, model 3….
Author Response
Reviewer #1.
- There is no criteria of inclusion and exclusion for patients` selection
Response: I really appreciate your comments. I modified the 1st paragraph in the Patient Selection part as follows.
“We selected subjects as inclusion criteria who were older than 20 years and had undergone health checkups in 2009 (n = 10,585,852). We monitored the subjects until 31 December 2018. As exclusion criteria, we set missing data (n = 746,403), previous cancer history (n = 153,456), and individuals who were diagnosed with cancer within a one-year lag period to minimize detection bias (n = 87,908). A final total of 9,598,085 subjects was included in this study.”
- The author should list a subhead: data collection with details about how they collect the data. Some definition for some variables were not given such as smoking, drinking, physical activity. The author should give specific standard
Response: I really appreciate your comments. I modified the end of 1st paragraph in the Patient Selection part as follows.
“Smoking criteria are divided into three classifications based on past and present smoking. Alcohol intake criteria are divided into three classifications based on frequency in 1 week and amount on one occasion (none; mild, < 30 g of alcohol/day; heavy, ≥ 30 g of alcohol/day). Physical activity level was collected using standardized self-reporting International Physical Activity Questionnaire.”
- I suggest the author may draw a restricted cubic spline that showed the relationship between eGFR and risk of HNSCC.
Response: I really appreciate your comments. I described this statement 2nd paragraph in Results part, and added Figure 1.
- Statistic analysis: the author should present more details for statistical method descriptions. How you deal with variable? How do you perform the analysis? How do you define the model 1, model 2, model 3.
Response: I really appreciate your comments. I modified the 2nd paragraph in Results part as follows.
“Statistical analyses were performed using SAS version 9.2 (SAS Institute, Cary, NC, USA). Basic characteristics are presented using descriptive analysis. Differences in baseline characteristics between groups were determined using Student’s t-test for continuous variables and the X2 test for categorical variables. HNC incidence was calculated by dividing the number of cases by 1,000 person-years. Cox proportional hazards models were applied to estimate hazard ratios (HRs) and 95% confidence intervals (CIs) for the associations between eGFR and risk of HNC. Subgroup analyses were performed by multivariable Cox proportional hazard models. Model 1 was unadjusted. Model 2 was adjusted for age, sex, income, smoking, alcohol consumption, and regular exercise. Model 3 was adjusted for age, sex, income, smoking, alcohol consumption, regular exercise, diabetes, and hypertension. A p-value < 0.05 was considered statistically significant.”
Reviewer 2 Report
The topic of the article is relevant and interesting as it tries to evaluate causal association of estimated glomerular filtration rate with a risk of head and neck cancer.
Questions/Comments
1. Is the research protocol approved by the Institutional Review Board consistent with the approval of the Research Ethics Committee? As a cohort study uses an individual information (and self-reported questionnaire was used to obtain information as well), ethical approval is obligatory.
2. In the introduction, the authors call age and gender as congenital factors. In epidemiology, risk factors such as age, gender, social and marital status, environmental factors, etc. are classified as modifiable and non-modifiable factors. Age and gender were used to be not modifiable risk factors. However, in our time this can be discussed. Nevertheless, term „congenital“ is a medical term usually referring to birth defects and diseases.
3. Clarification of the date until the cohort of the study was followed up is necessary, because the statement „to the date of diagnosis of HNC“ is not appropriate. If it was correct, all individuals would be with HNC at the end of the study.
4. Information on glomerular filtration rate as exposure variable is very limited. A method used to measure this index, time of exposure assessment is not given. At what stage of the study glomerular filtration rate was estimated? Was it the same during follow-up? What method was used to measure glomerular filtration?
5. According to the authors, only cancer patients were excluded from the study group. So, individuals, especially the old ones could have other diseases not only arterial hypertension and diabetes that were considered by the authors. Therefore, conclusion in the abstract is overestimated. I would suggest to delete a word „healthy“ and to list diseases that were taken into account. The same summarizing sentence is at the end of 1st paragraph on page 7 in the discussion part.
6. Table 2 provides information on hazard ratios (HR) of models 1-3, therefore HR1-HR3 should be in the top row of the table.
Author Response
Reviewer #2.
- Is the research protocol approved by the Institutional Review Board consistent with the approval of the Research Ethics Committee? As a cohort study uses an individual information (and self-reported questionnaire was used to obtain information as well), ethical approval is obligatory.
Response: I really appreciate your comments. I modified Institutional Review Board Statement at the end of Study population part.
The present study protocol was reviewed and approved by the Institutional Review Board (IRB)/Ethics Committee approval for the study was obtained from The Catholic University of Korea in accordance with the Declaration of Helsinki. [SC21ZISE0024]
- In the introduction, the authors call age and gender as congenital factors. In epidemiology, risk factors such as age, gender, social and marital status, environmental factors, etc. are classified as modifiable and non-modifiable factors. Age and gender were used to be not modifiable risk factors. However, in our time this can be discussed. Nevertheless, term “congenital” is a medical term usually referring to birth defects and diseases.
Response: I really appreciate your comments. The word “congenital” was replaced with the word “genetic” in the 1st sentence of the introduction part.
- Clarification of the date until the cohort of the study was followed up is necessary, because the statement, to the date of diagnosis of HNC is not appropriate. If it was correct, all individuals would be with HNC at the end of the study.
Response: I really appreciate your comments. I modified the 1st paragraph in the Patient Selection part as follows.
“We selected subjects as inclusion criteria who were older than 20 years and had undergone health checkups in 2009 (n = 10,585,852). We monitored the subjects until 31 December 2018. As exclusion criteria, we set missing data (n = 746,403), previous cancer history (n = 153,456), and individuals who were diagnosed with cancer within a one-year lag period to minimize detection bias (n = 87,908). A final total of 9,598,085 subjects was included in this study. Participants were defined as having HNC if they had admission records for HNC in their national health insurance data from 2010 to 2018.”
- Information on glomerular filtration rate as exposure variable is very limited. A method used to measure this index, time of exposure assessment is not given. At what stage of the study glomerular filtration rate was estimated? Was it the same during follow up? What method was used to measure glomerular filtration?
Response: I really appreciate your comments. This is an observational cohort study, however, estimated glomerular filtration rate (eGFR) was assessed one time in 2009. So I added a limitation at the end of last paragraph of Discussion part as follows.
Last, eGFR was measured only once in a health checkup in 2009. Dynamic changes of eGFR could affect the incidence of HNC.
- According to the authors, only cancer patients were excluded from the study group. So, individuals, especially the old ones could have other diseases not only arterial hypertension and diabetes that were considered by the authors. Therefore, conclusion in the abstract is overestimated. I would suggest to delete a word “healthy” and to list diseases that were taken into account. The same summarizing sentence is at the end of 1st paragraph on page 7 in the discussion part.
Response: I really appreciate your comments. I modified the Conclusion of Abstract part, at the end of 1st paragraph in the Discussion part, and at the last sentence of Conclusion part as follows, respectively.
Elevated eGFR was associated with the risk of some type of HNC, even in individuals with adjusted hypertension and diabetes without chronic diseases.
This finding emphasizes that people with adjusted hypertension and diabetes without chronic disease should be aware of the possibility of HNC if eGFR is 60 mL/min/1.73m2 or higher.
The present data provide a solid basis for future, larger studies aimed to assess whether eGFR screening of people with adjusted hypertension and diabetes may significantly decrease HNC incidence and mortality risk to justify the cost and effort.
6 Table 2 provides information on hazard ratios (HR) of models 1-3, therefore HR1-HR3 should be in the top row of the table.
Response: I really appreciate your comments. I added “hazard ratios and 95% confidence interval” on top row of the table 2.
Round 2
Reviewer 1 Report
No other comment.